# Dynamic Rank Reallocation for Module-Level LoRA

## Abstract

Low-Rank Adaptation (LoRA) and its variants have been widely applied in parameter-efficient fine-tuning of large language models. However, most existing approaches assign the same rank to all modules, which limits the representational capacity and adaptation flexibility of the finetuned model. Inspired by interference theory in human learning, which posits that forgetting outdated knowledge while allocating more resources to relevant new knowledge facilitates better learning, we propose the Dynamic rank re-Allocation method (DaRA). In DaRA, the rank of each module represents a direction in the parameter space, analogous to a knowledge component. During warm-up, DaRA allocates the same rank to all modules, allowing the model to form a coarse understanding of the task. Afterwards, DaRA reallocates ranks by discarding less useful directions from unimportant modules (outdated knowledge) and assigning more ranks to important modules (new knowledge). This dynamic adjustment mirrors the human learning process described by interference theory. Extensive experiments across diverse model architectures and downstream tasks demonstrate that DaRA consistently outperforms existing baselines.

## 1 Introduction

In recent years, the emergence of large-scale pre-trained models on general-domain corpora (Brown et al., 2020; OpenAI, 2023; Touvron et al., 2023a;b; Bai et al., 2023; Mesnard et al., 2024; Zeng et al., 2023) has firmly established the pretrain–fine-tune paradigm as the dominant framework for natural language processing and multimodal learning. Alongside this trend, parameter-efficient fine-tuning (PEFT) methods (Li & Liang, 2021; Liu et al., 2022; Houlsby et al., 2019; Hu et al., 2022) have emerged, enabling models to adapt to downstream tasks by updating only a small fraction of parameters while keeping the majority frozen. Among various PEFT methods, Low-Rank Adaptation (LoRA) (Hu et al., 2022) was proposed early and has become one of the most influential approaches. LoRA is based on the intuition that parameter updates during fine-tuning often lie in a low-dimensional subspace. It reparameterizes the update matrices as the product of two low-rank matrices, thereby drastically reducing the number of trainable parameters while maintaining performance comparable to full fine-tuning. Thanks to its simplicity, efficiency, and architecture-free design, LoRA has been widely adopted in diverse domains, including language modeling (Hu et al., 2022), multimodal learning (Microsoft et al., 2025), and recommender systems (Zhu et al., 2024), and has quickly become one of the most cited PEFT methods.

Most existing approaches allocate the same rank to all modules, which restricts the representational capacity and adaptation flexibility of the finetuned model. From our observations during full-parameter fine-tuning and LoRA, not only do different modules within the same model exhibit different optimal ranks, but even the same model may require different ranks when trained on different datasets. This suggests that it is unreasonable to predefine a fixed rank allocation for each module in advance. A more practical strategy is to start with an initial allocation of ranks, and then dynamically reallocate them during training. Although some prior works have considered rank variation (Zhang et al., 2023b;a), they only explore one-sided changes—either rank reduction or rank increase—rather than true reallocation across modules. Moreover, these studies do not explicitly target the problem of redistributing ranks among modules. Rank-decreasing methods, inspired by pruning, primarily aim to lower the number of parameters while retaining performance, but they ignore the potential benefit of reallocating ranks and are strictly constrained by the upper bound of

the initial rank. Conversely, rank-increasing methods provide module-specific adjustments, but their training process is fundamentally limited by the initially low-rank matrices, often leading to training instability.

To address the above limitations, we draw inspiration from interference theory in human learning and propose a dynamic rank reallocation method (DaRA). Interference theory suggests that when learning a new task, humans must first suppress or forget useless prior knowledge before effectively acquiring new knowledge. Analogously, in DaRA, the rank of each module is viewed as a direction in the parameter space. During the warm-up stage, DaRA first allocates the same rank to all modules, enabling the model to form a coarse understanding of the task. It then reallocates ranks by discarding less important directions from unimportant modules (forgetting outdated knowledge) and allocating new directions to critical modules to capture more task-relevant information (acquiring new knowledge). To make this process efficient, we introduce a lightweight metric (Diao et al., 2023) to assess the relative importance of modules and guide the reallocation. This method mirrors the human learning mechanism of forgetting and reinforcing, thus improving both adaptability and performance.

Our contributions are summarized as follows:

- Motivated by interference theory in human learning and empirical observations from full-parameter fine-tuning, we propose **DaRA**, a dynamic rank reallocation method. DaRA unifies rank decrease and rank increase, enabling modules to first share a uniform rank during warm-up and then adaptively reallocate ranks.
- We introduce a lightweight metric to efficiently assess module importance and guide the reallocation process, enabling effective parameter reallocation without additional gradient computation.
- Through extensive experiments across multiple architectures and downstream tasks, we demonstrate that **DaRA achieves consistent improvements over existing PEFT baselines**, offering superior adaptability and performance under the same parameter budgets.

## 2 RELATED WORK

**Parameter-Efficient Fine-Tuning (PEFT)** (Li & Liang, 2021; Liu et al., 2022; Houlsby et al., 2019; Hu et al., 2022) has gained increasing attention as a way to reduce the computational and storage costs of adapting large pre-trained models. Existing PEFT methods can be broadly categorized into selective and additive approaches. Selective methods update only a subset of the original model parameters, such as BitFit (Zaken et al., 2022), which tunes only bias terms, or Fish-Mask (Sung et al., 2021), which selects parameters based on Fisher information, thereby reducing trainable parameters but sometimes at the cost of sub-optimal performance. Additive methods, including Adapters (Houlsby et al., 2019), LoRA (Hu et al., 2022), Prefix-Tuning (Li & Liang, 2021), and related variants, introduce small trainable modules into the model, allowing efficient fine-tuning without additional inference overhead. Hybrid approaches (Mao et al., 2022; Lawton et al., 2023) further combine these strategies, often incorporating parameter sharing or pruning, to exploit redundancy in PEFT modules and achieve better performance under limited parameter budgets. Overall, PEFT enables flexible and efficient adaptation of large models while maintaining competitive accuracy with significantly fewer trainable parameters.

**Low-Rank Adaptation (LoRA)** (Hu et al., 2022) belongs to the second category of PEFT methods, i.e., additive approaches, which introduce low-rank trainable modules into the model for efficient fine-tuning without adding inference overhead. LoRA and its variants have received widespread attention and inspired various research directions to improve their expressive capacity and adaptability. For instance, Zhang et al. (2023b) employ SVD decomposition and prune less significant singular values for more efficient updates; Hyeon-Woo et al. (2022) focus on low-rank Hadamard products in federated learning; Liu et al. (2024) utilize orthogonal factorization when fine-tuning diffusion models; Renduchintala et al. (2024) leverage weight tying to further reduce trainable parameters; Yeh et al. (2024) introduce a unified LoRA family framework for Stable Diffusion; Ponti et al. (2023) select different combinations of LoRA modules via a routing function for different tasks; and Kopiczko et al. (2024) implement learnable scaling vectors to adjust shared pairs of frozen random matrices across layers.

Among these methods, some specifically focus on enhancing the rank of LoRA modules to improve expressiveness. For example, KronA (Edalati et al., 2022) increases the module rank via Kronecker

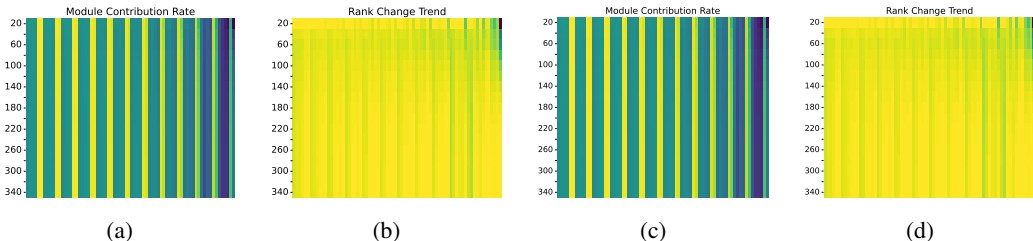

Figure 1: **Module Rank Contribution and Rank Change Trend.** The subplots (a) and (b) show the module rank of the OPT-125M(Zhang et al., 2022) model on the Winogrande (Sakaguchi et al., 2020) dataset and the normalization across different checkpoints for the same module, respectively, while (c) and (d) correspond to the results of DeBERTaV3-base (He et al., 2023) on the CoLA (Wang et al., 2019) dataset.

products, and Efficient Learning with Sine-Activated Low-Rank Matrices (Ji et al., 2025) boosts rank using sine-based activations. These approaches generally increase the rank uniformly across all modules, lacking flexible allocation for different modules. To address this, some methods explore dynamic rank adjustment: AdaLoRA (Zhang et al., 2023b) reduces ranks based on module importance to save parameters, while IncreLoRA (Zhang et al., 2023a) increases ranks to achieve module-specific allocation. However, previous methods still suffer from certain limitations: approaches that uniformly increase ranks across all modules ignore the heterogeneous importance of different modules, while methods that only decrease or only increase ranks restrict the flexibility and expressiveness of LoRA. To overcome these issues, our work introduces dynamic rank reallocation (DaRA), which is more flexible and expressive than single-direction rank adjustment methods. Moreover, DaRA can be combined with strategies that uniformly increase ranks across all modules to further enhance parameter-efficient fine-tuning.

## 3 ANALYSIS OF FULL-PARAMETER FINE-TUNING AND LoRA

To further demonstrate the effectiveness of our method and the rationale for allocating different ranks to different modules, we conduct experiments under both full-parameter fine-tuning (FFT) and LoRA-based fine-tuning.

### 3.1 RANK DYNAMICS IN FULL-PARAMETER FINE-TUNING

**FFT Formulation:** Let $\mathcal{M}$ denote a pretrained model with parameter set $\Theta = \{W_1, W_2, \ldots, W_n\}$. Fine-tuning (FT) refers to the process of adapting $\mathcal{M}$ on a downstream dataset by updating a subset of parameters $\Theta' \subseteq \Theta$. In particular, when $\Theta' = \Theta$, i.e., all parameters are updated, the process is referred to as full-parameter fine-tuning (FFT).

To better understand the learning patterns of different modules under FFT, we conduct experiments on multiple models across various datasets to trace the rank evolution of each module. To achieve this, we measure the effective rank of module updates during training as follows.

**Analysis Method:** Let $W_i^0 \in R^{d \times k}$ denote the pretrained weight in set $\Theta$ and $W_i^t$ denote its weight after the $t$-th training step. The module-wise update is computed as:

$$\Delta W_i^t = W_i^t - W_i^0 \tag{1}$$

To find the effective rank of each weight, we apply singular value decomposition (SVD) on $\Delta W_i^t$:

$$\Delta W_i^t = U_i^t \Sigma_i^t (V_i^t)^\top \tag{2}$$

where $\Sigma_i^t = \text{diag}(\sigma_1, \sigma_2, \ldots, \sigma_{\min(d,k)})$ contains the singular values. Given a threshold $\tau$, the effective rank $r^t$ of the module at step $t$ is defined as the number of singular values whose normalized contribution exceeds $\tau$:

$$r_i^t = \sum_{l=1}^{\min(d,k)} \mathbf{1}\left(\frac{\sigma_l}{\sum_j \sigma_j} \geq \tau\right) \tag{3}$$

where $\mathbf{1}(\cdot)$ is the indicator function. This measure captures the number of dominant directions contributing significantly to the module's weight update at each training step.

**Results and Analysis:** The rank $r^t$ of different modules across training steps is visualized in Figure 1. In Figure 1 (b) and Figure 1 (d), we show how the ranks of different modules evolve over the course of training, while Figure 1 (a) and Figure 1 (c) present a normalized visualization of ranks for all modules at a single checkpoint, mapped to the range $[0, 1]$. Each row corresponds to a specific checkpoint, and each column represents a particular module. The results reveal significant differences in rank dynamics among modules: some modules exhibit gradually increasing rank during training, while others remain rank or slightly decrease. This non-uniform distribution of rank dynamics indicates that different modules contribute unevenly to the overall model expressiveness during FT, providing valuable insights for subsequent adaptive rank allocation strategies in LoRA-based PEFT methods.

## 3.2 MODULE IMPORTANCE OF LORA

**LoRA Formulation:** LoRA models the weight update $\Delta W_i$ of a pre-trained weight matrix $W_i^0 \in \mathbb{R}^{d \times k}$ as a product of two low-rank matrices $B \in \mathbb{R}^{d \times r}$ and $A \in \mathbb{R}^{r \times k}$, i.e.,

$$W_i' = W_i^0 + \Delta W_i = W_i^0 + BA \tag{4}$$

where $r \ll \min(d, k)$ is the rank. In standard practice, the same rank $r$ is allocated uniformly to all LoRA modules in the model.

**Experimental Setup:** To investigate the effect of module-specific rank allocation, we conducted experiments on the OPT model under a fixed total rank budget and varied which modules were fine-tuned. The evaluation was performed on SST-2 and Winogrande. The modules we examined include the query (Q), key (K), value (V), and output (O) projections in the attention mechanism, as well as the two fully connected layers in the FNN (FC1, FC2), along with their combinations. This setup represents an extreme form of rank allocation, where some modules are assigned a rank of 0 while others receive allocated ranks.

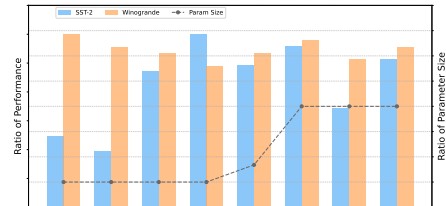

Figure 2: **Parameter Size vs. Performance between Different Modules.** Tuning different modules results in different performance.

**Results and Analysis:** Our experiments show that in Figure 2, the optimal modules to fine-tune vary across different models and datasets. In some cases, fine-tuning only the query and value projections can achieve high performance with relatively low overhead, while in other cases, fine-tuning only the projection layers yields the best results with minimal parameter cost. Moreover, the effectiveness of fine-tuning FC1 and FC2 differs across datasets. These results highlight that uniform rank allocation across all modules may not be optimal, and that considering module-specific rank allocation is crucial for maximizing the effectiveness of LoRA under a limited total rank budget.

## 4 METHOD

As observed in section 3, the optimal rank of different modules varies across models and datasets, making it difficult to predefine the optimal rank for each module. A representative example is LoRA, which follows a predefined allocation strategy and assumes that all modules share the same optimal rank. This assumption is clearly unrealistic. To address this, we propose a compromise: we first allocate the same initial rank to all modules, and then reallocate ranks during training based on the behavior of the corresponding parameters. The distinction between the predefined allocation paradigm and our reallocation paradigm is illustrated in Figure 3.

Figure 3: **Differences between DaRA and Allocation LoRA, and the Analogy to Human Learning.** The left figure illustrates the difference between DaRA and predefined rank allocation, showing that this is only one component of the DaRA method, while the right figure presents the analogy between the method and human learning.

## 4.1 RANK ALLOCATION

At the beginning of training, our method allocates the same initial rank to all modules, following the common practice in existing LoRA-based approaches. This uniform allocation ensures a fair starting point where each module is granted equal expressive capacity, avoiding premature bias toward certain directions. During this warm-up phase, the model explores the parameter space under balanced conditions before adaptive reallocation begins.

LoRA can be equivalently formulated by representing $A$ and $B$ as collections of vectors:

$$A = [a_1, a_2, \ldots, a_r], \quad B = [b_1, b_2, \ldots, b_r], \tag{5}$$

with $a_i \in \mathbb{R}^{d_{in}}, ; b_i \in \mathbb{R}^{d_{out}}$. Equation 4 can then be further expanded as:

$$W = W^{(0)} + w_1 + w_2 + \cdots + w_r = W^{(0)} + b_1^T a_1 + b_2^T a_2 + \cdots + b_r^T a_r, \tag{6}$$

where $w_i$ is a rank-1 matrix obtained by the outer product of $b_i$ and $a_i$.

For the subsequent reallocation of ranks, we introduce a learnable scaling factor $\lambda_i$ for each component $w_i$:

$$W = W^{(0)} + \sum_{i=1}^{r} \lambda_i w_i = W^{(0)} + \sum_{i=1}^{r} \lambda_i b_i^T a_i, \tag{7}$$

where each $\lambda_i$ is updated via backpropagation. The sequence $\lambda_i$ reflects the relative importance of different directions, allowing the model to adjust capacity dynamically by emphasizing or suppressing specific rank components. In this way, although all modules start with identical ranks, their effective contributions gradually diverge as $\lambda_i$ values evolve, leading to a more adaptive and fine-grained parameter re-allocation. To make this formulation more compact, we rewrite the update in matrix form. Let $\Lambda = \text{diag}(\lambda_1, \lambda_2, \ldots, \lambda_r)$, then the update can be written compactly as:

$$\Delta W = B \Lambda A. \tag{8}$$

To further ensure that each component $w_i$ captures distinct directions in the parameter space, we impose an orthogonality regularization on $A$ and $B$:

$$R(A, B) = ||A^\top A - I||_F^2 + ||B^\top B - I||_F^2, \tag{9}$$

where $I$ is the identity matrix. This reduces dependencies between rank components and ensures that $\lambda_i$ modulates contributions of diverse and independent directions—analogous to the orthogonal bases $U, V$ in singular value decomposition (SVD).

In practice, this SVD-inspired triplet structure $(B, \Lambda, A)$ replaces the original LoRA matrix, with $\Lambda$ stored as a one-dimensional tensor, introducing only $r$ additional parameters per module. This design keeps the formulation compact while enabling more expressive and flexible adaptation.

## 4.2 RANK REALLOCATION

To determine which modules have redundant ranks that should be pruned and which modules are important and deserve more rank allocation, we introduce the PQ Index (PQI) from the pruning literature, a lightweight metric for measuring parameter sparsity. The definition is as follows:

**Definition 1 (PQ Index).** For any $0 < p \leq 1 < q$ and non-zero $w \in \mathbb{R}^d$, the PQ Index is

$$I_{p,q}(w) = d^{\frac{1}{q} - \frac{1}{p}} \frac{\|w\|_p}{\|w\|_q}, \qquad \|w\|_p = \Big( \sum_{i=1}^{d} |w_i|^p \Big)^{1/p}. \tag{10}$$

We write $I(w)$ when $(p, q)$ are clear from context. Intuitively, larger $I(w)$ indicates a denser (less sparse) distribution of mass across coordinates, whereas smaller $I(w)$ indicates concentration on a few entries (higher sparsity).

We employ PQI on the diagonal of matrix $\Lambda$ in Equation 8 to quantify the relative importance of each module. A larger PQI, which arises when the values of $\Lambda$ are relatively uniform, suggests that all ranks (i.e., the knowledge directions discussed in the previous subsection) contribute meaningfully, indicating the potential benefit of allocating additional ranks to capture richer parameter directions. Conversely, when certain entries of $\Lambda$ are substantially smaller than others, the resulting lower PQI reflects redundancy within the module, implying that some directions are uninformative and the effective rank of the module can be reduced accordingly. In addition, PQI also satisfies the following property (proof can be found in Appendix B), which allows us to determine how much the rank of each module should be decreased.

**Theorem 1.** Let $M_r$ denote the set of $r$ indices of $w$ with the largest magnitudes, and let $\eta_r$ be the smallest value such that

$$\sum_{i \notin M_r} |w_i|^p \leq \eta_r \sum_{i \in M_r} |w_i|^p. \tag{11}$$

Then

$$r \geq d \left(1 + \eta_r\right)^{-\frac{q}{q-p}} \left[ I(w) \right]^{\frac{qp}{q-p}}. \tag{12}$$

After the warm-up phase, during which the model fully explores the parameter space under uniformly allocated ranks, we employ PQI to guide dynamic rank adjustment. This design ensures that the model begins adaptation only after establishing a balanced foundation across all modules.

Concretely, by applying Equation 12 with a given tolerance $\eta$, we can determine the precise number of ranks to be reduced for each module. Modules with lower PQI are considered to contain redundant or uninformative directions, and their ranks are therefore safely reduced. This process parallels the role of forgetting in human learning, where discarding useless or conflicting information enables more efficient acquisition of new knowledge and frees capacity for more meaningful directions.

Furthermore, to ensure that important modules continue to gain sufficient expressive power, we perform rank expansion at regular intervals of $\nu$ steps. Specifically, modules with the highest PQI are assigned additional ranks, allowing them to capture richer parameter directions.

Through PQI-guided rank reallocation, DaRA not only eliminates redundant directions but also strengthens the most informative ones, thereby improving both training efficiency and parameter utilization. The detailed algorithmic procedure is provided in Appendix A for completeness.

## 5 EXPERIMENT

We compare our method on natural language understanding tasks using DeBERTaV3-base (He et al., 2023) with an additional classification layer on Glue (Wang et al., 2019). Furthermore, we investigate the performance of larger language models, including OPT-125M, Zhang et al. (2022) Qwen1.5-1.8B (Bai et al., 2023), and LLaMA3.1-8B (Dubey et al., 2024), on question answering benchmarks such as MathQA (Amini et al., 2019), OpenBookQA (Mihaylov et al., 2018), Winogrande (Sakaguchi et al., 2020), and BoolQ (Clark et al., 2019).

**Implementation Details.** We implement all algorithms in PyTorch (Paszke et al., 2019), based on the publicly available Huggingface Transformers library (Wolf et al., 2020). LoRA scales $\Delta x$ by $\alpha/r$, where we set $\alpha = 2r$, so that the output magnitude remains consistent across different values of $r$. This reduces the need to retune the learning rate when varying $r$. Following LoRA, we apply the same scaling for Eq. (3) and fix $\alpha$ as $2r$.

Table 1: **Comparison between Methods on Glue.**

| Method | Parmas | MNLI Acc | SST-2 Acc | COLA Acc | QQP Acc | QNLI Acc | AVERAGE |
|---|---|---|---|---|---|---|---|
| LoRA | 1.33M | 86.85 | 92.35 | 85.05 | 88.34 | 90.48 | 88.61 |
| AdaLoRA | 1.31M | 88.42 | 92.30 | 85.19 | 88.89 | 91.45 | 89.25 |
| IncreLoRA | 1.36M | 88.51 | 92.79 | 85.33 | 88.92 | 90.77 | 89.26 |
| DaRA$_{raw}$ | 1.34M | **89.22** | 92.89 | **86.35** | **89.08** | 91.41 | **89.79** |
| DaRA$_{task}$ | 1.34M | 89.14 | **93.16** | 85.82 | 88.97 | **91.49** | 89.72 |

**Methods.** The baseline and our method are described as follows:

- **LoRA** (Hu et al., 2022) is a state-of-the-art method for parameter-efficient fine-tuning. It parameterizes incremental updates using two low-rank matrices, with the number of trainable parameters determined by the rank $r$ and the number of adapted weight matrices $n$.
- **AdaLoRA** (Zhang et al., 2023b) adaptively allocates parameter ranks during training based on importance scores, thereby improving parameter utilization and efficiency.
- **IncreLoRA** (Zhang et al., 2023a) gradually increases the rank of LoRA modules during training by dynamically expanding the parameter capacity.
- **DaRA** has been introduced in section 4. For rank expansion, we consider two initialization schemes: (1) the LoRA default (Hu et al., 2022), with $B = 0$, $A$ random, and $\Lambda$ is an identity matrix; (2) task-direction initialization, using the mean of the top-$k$ principal directions from the previous rank.

## 5.1 NATURAL LANGUAGE UNDERSTANDING

**Models and Datasets.** We evaluate the fine-tuning performance of DeBERTaV3-base (He et al., 2023) using the proposed algorithm. Experiments are conducted on the General Language Understanding Evaluation (GLUE, (Wang et al., 2019)) benchmark. Specifically, we select five representative datasets from GLUE for training, including MNLI, CoLA, QNLI, SST-2, and QQP. Dataset details are summarized in Appendix C.

**Implementation Details.** We use PyTorch (Paszke et al., 2019) and Transformers (Wolf et al., 2020), and run all experiments on NVIDIA 4090 GPUs. IncreLoRA is applied to fine-tune DeBERTaV3-base (He et al., 2023), which has 12 layers, hidden size 768, and 183M parameters. Update matrices are applied to all backbone weights, and the trainable parameter count is determined by the final total rank $r_{final}$. For instance, $r_{avg} = 2$ yields about 0.32M trainable parameters, while we set $r_{avg} = 8$ in our experiments. Due to varying module sizes and early-stopping checkpoints of our method, the actual parameter count may differ from this budget.

**Results.** In Table 1, we compare the proposed method with the baseline models. Since the allocation of trainable parameters for both the baseline methods and our method is not fixed across different tasks, we report the average number of parameters across all tasks. The experimental results show that, under different tasks and parameter budgets, the parameter size of our method falls between AdaLoRA and IncreLoRA, and its performance exhibits improvements compared to both.

Further analysis reveals a consistent trend in parameter allocation: our method tends to reduce the rank of the QKV modules in LoRA, while increasing the rank of the projection layers and FNN modules. This observation is consistent with the conclusions of IncreLoRA as well as the heatmap results in subsection 3.1. Drawing upon findings from the knowledge editing literature, it can be inferred that the QKV layers primarily capture general relational information, whereas the projection layers and FNN modules are more likely to encode domain-specific knowledge. Since such relational information is relatively universal across domains and has already been sufficiently learned during pre-training, allocating more parameter budget to the projection and FNN modules during fine-tuning can more effectively enhance model performance. Although the FNN modules have far more parameters than the attention modules, as shown in the table, our method still demonstrates superior performance under nearly the same parameter scale.

Table 2: **Comparison of Time Between Methods Across Different Base Models.**

| Base Model | Method | MathQA Acc | BoolQA Acc | OpenbookQA Acc | Winogrande Acc | Average |
|---|---|---|---|---|---|---|
| | LoRA | 22.02 | 62.20 | 16.06 | 50.81 | 37.77 |
| | AdaLoRA | 22.47 | 62.20 | 16.11 | 51.19 | 37.99 |
| OPT-125M ($r_{\mathrm{avg}} = 16$) | IncreLoRA | 22.56 | 62.20 | **16.21** | 51.51 | 38.12 |
| | DaRA$_{\mathrm{raw}}$ | 22.42 | **62.21** | 16.12 | 51.41 | 38.04 |
| | DaRA$_{\mathrm{task}}$ | **22.71** | 62.20 | 16.18 | **51.70** | **38.20** |
| | LoRA | 25.33 | 69.17 | 22.30 | 51.49 | 42.07 |
| | AdaLoRA | 26.62 | 69.17 | 25.83 | 51.57 | 43.30 |
| Qwen1.5-1.8B ($r_{\mathrm{avg}} = 8$) | IncreLoRA | 26.70 | 69.20 | 25.59 | 51.64 | 43.28 |
| | DaRA$_{\mathrm{raw}}$ | 26.73 | **69.31** | **25.94** | **51.64** | **43.41** |
| | DaRA$_{\mathrm{task}}$ | **26.78** | 69.20 | 25.61 | 51.62 | 43.25 |
| | LoRA | 39.42 | 84.35 | 37.23 | 55.44 | 54.11 |
| | AdaLoRA | 39.56 | 84.54 | 37.96 | 55.57 | 54.41 |
| LLaMA3.1-8B ($r_{\mathrm{avg}} = 8$) | IncreLoRA | 39.62 | 84.65 | 38.23 | **55.69** | 54.55 |
| | DaRA$_{\mathrm{raw}}$ | 39.65 | **84.91** | **38.40** | 55.60 | **54.64** |
| | DaRA$_{\mathrm{task}}$ | **39.68** | 84.68 | 38.08 | 55.66 | 54.53 |

## 5.2 REASONING AND QUESTION ANSWER

**Models and Datasets.** We evaluate the proposed algorithm on larger language models, including OPT-125M, Qwen-1.8B, and LLaMA3.1-8B, using four question answering benchmarks: MathQA (Amini et al., 2019) for math word problems, OpenBookQA (Mihaylov et al., 2018) for scientific reasoning, Winogrande (Sakaguchi et al., 2020) for commonsense reasoning, and BoolQ (Clark et al., 2019) for yes/no questions from search queries. Additional dataset details are given in Appendix C.

**Implementation Details.** We conduct our experiments on NVIDIA A800 GPUs. We adopt IncreLoRA with different target average ranks for different backbone models. For OPT-125M (Zhang et al., 2022), we set the average target rank to $r_{\mathrm{avg}} = 16$ and apply LoRA update matrices to all weight matrices in the network. For Qwen (Bai et al., 2023) and LLaMA (Dubey et al., 2024), we use $r_{\mathrm{avg}} = 8$ and restrict LoRA updates to the attention projection matrices $Q, K, V, O$.

**Results.** We compare the proposed method with the baseline approaches in Table 2. The experimental results show that on OPT-125M, our method consistently outperforms the baseline models, and the improvements are similar to those observed on DeBERTaV3-base, demonstrating the effectiveness of the method on both encoder- and decoder-style models. For Qwen and LLaMA, the parameter sizes of the LoRA-applied Q, K, V, and O weights are nearly identical. Nevertheless, our method still achieves the best overall performance, indicating that the adaptive allocation strategy remains effective even when the underlying parameter budgets are highly similar across modules.

## 6 ANALYSIS

### 6.1 EXPLANATION FOR PQI

As stated in subsection 4.2, PQI reflects parameter sparsity, and thus, when computing $\Lambda$ in the formula, it indicates the importance of the current set of directions. Moreover, this metric requires significantly less computation than the importance scores used in AdaLoRA and IncreLoRA.

Furthermore, we conduct experiments on multiple datasets using the same model with randomly initialized ranks for different modules, without performing

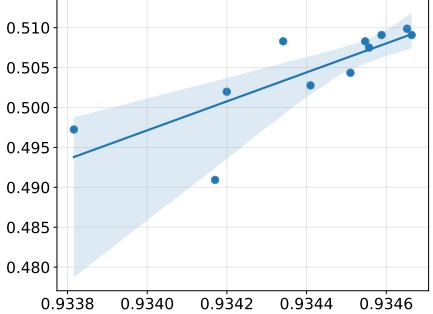

Figure 4: **PQI vs Test Accuracy.** It can be observed that PQI and test accuracy exhibit a positive correlation.

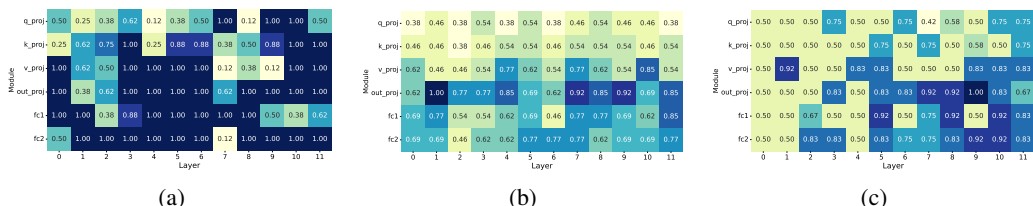

(a)                 (b)                 (c)

Figure 5: **Rank Distribution with Different Methods.** From left to right are the rank distributions across different layers and modules for AdaLoRA, IncreLoRA, and our method.

any rank reallocation, and directly fine-tune the model. We observe from Figure 4 that during the warm-up phase, i.e., when the initial accuracy is rising, PQI is positively correlated with test performance. This also provides additional explanation for the effectiveness of using the PQI metric to perform rank reallocation.

## 6.2 RANK DISTRIBUTION

Similar to AdaLoRA and IncreLoRA, we also analyze the rank distributions across different layers and modules. Figure 5 presents the rank distributions of three methods on the Winogrande dataset using the OPT model under approximately the same parameter budget. We find that AdaLoRA produces a more evenly spread distribution, where ranks are allocated across different layers and modules. In contrast, IncreLoRA focuses more heavily on the FNN modules and the projection layers of the attention modules, while largely ignoring the functional differences across layers.

Our method lies between the two: it emphasizes the FNN and projection layers while also allocating more ranks to the QKV modules in later layers. Since these layers directly affect task performance and QKV modules are key to capturing attention, prioritizing them improves expressiveness, particularly for reasoning and semantic understanding tasks where later-layer attention strongly influences output correctness.

Moreover, since our distribution is more concentrated than AdaLoRA but not as extreme as IncreLoRA, the parameter count of our method naturally falls between the two.

## 6.3 CONVERGENCE SPEED AND STABILITY

We present the convergence curves of AdaLoRA, IncreLoRA, and our method. It can be observed in Figure 6 that our method exhibits a convergence speed comparable to AdaLoRA in the early stage and significantly faster than IncreLoRA. In the later stage, however, the loss of our method becomes comparable to that of IncreLoRA and higher than that of AdaLoRA.

This is because, in the early training stage, both our method and AdaLoRA start with a higher rank, enabling faster convergence than IncreLoRA. With a higher initial rank, our method is more likely to maintain an advantage and achieve lower loss. Compared to AdaLoRA, both methods show similar convergence

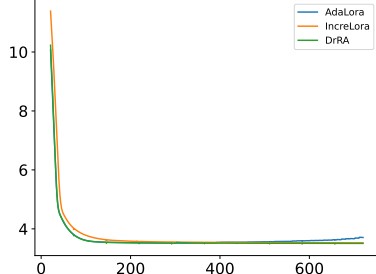

Figure 6: **Convergence Speeds of Different Methods.**

speed before rank reduction, but later our method gradually increases the rank, providing more parameter directions and thus sustaining or further reducing the loss. In contrast, AdaLoRA either keeps or reduces the rank, which may cause the loss to stagnate or even increase.

## 7 CONCLUSION

In summary, we propose DaRA, a module-level dynamic rank reallocation method that integrates previous rank reduction and rank expansion approaches while combining the advantages of both. Moreover, it is the first to introduce the PQI metric, which significantly reduces the computational overhead associated with module selection.

## ETHICS STATEMENT

This work does not raise any direct ethical concerns. All experiments are conducted on publicly available datasets and do not involve human subjects, private information, or sensitive content. The proposed methods are intended for advancing research on parameter-efficient fine-tuning of large language models. Potential societal impacts are consistent with those of general LLM research, including both positive applications (e.g., lowering computational cost and energy consumption) and risks of misuse (e.g., generating harmful or misleading text). We encourage responsible use of our methods and adherence to ethical guidelines in AI research and deployment.

## REPRODUCIBILITY STATEMENT

We have taken several measures to ensure the reproducibility of our work. All datasets used in this paper are publicly available and properly cited. The experimental settings, including model architectures, hyperparameters, training steps, and evaluation metrics, are described in detail in the main text and Appendix. For key results, we report the average performance over multiple random seeds. In addition, we provide pseudo-code and algorithmic descriptions in the appendix for clarity. Source code and instructions for reproducing our experiments will be made available upon publication.

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

## A  METHOD ALGORITHM

---

**Algorithm 1** DaRA. $\mathcal{D}$ is dataset, $\mathcal{M}$ is target module set, $\mathcal{A}, \mathcal{B}, \mathcal{E}$ is the set of parameter matrices, $T$ is total steps, $\mathcal{W}$ is warmup steps, $\eta$ is learning rate, $h$ is the number of modules selected in each round, $r^{final}$ is final total rank, interval $\mu$.

---

**Input**:$\mathcal{D}, T, \mathcal{W}, \eta, h, r^{final}$
**Parameter**: $\mathcal{A}, \mathcal{B}, \mathcal{E}$
 1: **for** $M_k$ in $\mathcal{M}$ **do**
 2:     initalization$(a_k, b_k, \lambda_k)$
 3: **end for**
 4: **for** $t = 1, ..., T$ **do**
 5:     Compute $I_{M_k}^{(t)}$ **for** $M_k$ **in** $\mathcal{M}$
 6:     **if** $t = \mathcal{W}$ **then**
 7:         **for** $\hat{I}_k^{(t)}$ in min$-h$ $\{\hat{I}_1^{(t)}, ..., \hat{I}_n^{(t)}\}$ **do**
 8:             Compute rank to decrease $r_{\text{decrease}}$ according $\hat{I}_k^{(t)}$
 9:             Delete smallest $r_{\text{decrease}}$ $\lambda_k$ and relative $a_k, b_k$
 10:             Update $r_{\text{total}}$
 11:         **end for**
 12:     **end if**
 13:     **if** $t > \mathcal{W}$ and $r_{\text{total}} < r_{\text{final}}$ **then**
 14:         **if** $t \% \mu == 0$ **then**
 15:             **for** $\hat{S}_k^{(t)}$ in top$-h$ $\{\hat{S}_1^{(t)}, ..., \hat{S}_n^{(t)}\}$ **do**
 16:                 Append new $a_k, b_k, \lambda_k$ to $A_k^{(t)}, B_k^{(t)}, \Lambda_k^{(t)}$
 17:                 initalization$(a_k, b_k, \lambda_k)$
 18:             **end for**
 19:         **end if**
 20:     **end if**
 21:     Update $\mathcal{A}^{(t)}, \mathcal{B}^{(t)}, \mathcal{E}^{(t)}$
 22: **end for**
**Output**: The fine-tuned parameters $\{\mathcal{A}^{(T)}, \mathcal{B}^{(T)}, \mathcal{E}^{(T)}\}$

---

## B  THEORETICAL PROOF

*Proof of Theorem 2.* Recall that $M_r$ is the largest $r$ components of $w$, and $\eta_r$ is a constant such that

$$\sum_{i \notin M_r} |w_i|^p \leq \eta_r \sum_{i \in M_r} |w_i|^p.$$

Therefore,

$$\|w\|_p = \left( \sum_{1 \leq i \leq d} |w_i|^p \right)^{\frac{1}{p}} = \left( \sum_{i \in M_r} |w_i|^p + \sum_{i \notin M_r} |w_i|^p \right)^{\frac{1}{p}}$$

$$\leq \left( \sum_{i \in M_r} |w_i|^p + \eta_r \sum_{i \in M_r} |w_i|^p \right)^{\frac{1}{p}} = \left( \sum_{i \in M_r} |w_i|^p \right)^{\frac{1}{p}} (1 + \eta_r)^{\frac{1}{p}}$$

$$\leq \left( \sum_{i \in M_r} |w_i|^q \right)^{\frac{1}{q}} r^{\frac{1}{p} - \frac{1}{q}} (1 + \eta_r)^{\frac{1}{p}} = \|w\|_q\, r^{\frac{1}{p} - \frac{1}{q}} (1 + \eta_r)^{\frac{1}{p}}.$$

Rearranging the above inequality gives

$$r \geq d(1 + \eta_r)^{-\frac{q}{q-p}} \left[ I(w) \right]^{\frac{qp}{q-p}}.$$

$\square$

## C   DETAILS OF DATASET

### C.1   DETAILS OF GLUE

We provide a detailed description of GLUE and five selected subtasks below.

**GLUE (General Language Understanding Evaluation)** is a comprehensive benchmark for natural language understanding, designed to assess the generalization ability of models across a diverse set of tasks. It comprises nine subtasks in total, spanning grammatical acceptability, sentiment analysis, semantic similarity, paraphrase detection, natural language inference, and question–answer matching.

- **CoLA (Corpus of Linguistic Acceptability)**: A single-sentence classification task that evaluates whether a sentence is grammatically acceptable.
- **SST-2 (Stanford Sentiment Treebank)**: A single-sentence sentiment classification task that determines whether a movie review is positive or negative.
- **QQP (Quora Question Pairs)**: A sentence-pair classification task that identifies whether two questions from Quora convey the same meaning.
- **MNLI (Multi-Genre Natural Language Inference)**: A natural language inference task that predicts whether a hypothesis is entailed by, contradicts, or is neutral with respect to a given premise.
- **QNLI (Question Natural Language Inference)**: A sentence-pair classification task derived from question answering, which determines whether a given sentence contains the answer to a question.

Table 3 presents the relevant statistics for each task.

Table 3: Summary of the GLUE benchmark.

| Corpus | Task | #Train | #Dev | #Test | #Label |
|--------|------|--------|------|-------|--------|
| *Single-Sentence Classification (GLUE)* | | | | | |
| CoLA | Acceptability | 8.5k | 1k | 1k | 2 |
| SST-2 | Sentiment | 67k | 872 | 1.8k | 2 |
| *Pairwise Text Classification (GLUE)* | | | | | |
| MNLI | NLI | 393k | 20k | 20k | 3 |
| QQP | Paraphrase | 364k | 40k | 391k | 2 |
| QNLI | QA/NLI | 108k | 5.7k | 5.7k | 2 |

### C.2   DETAILS OF DATASET FOR QUESTION ANSWERING

We provide detailed introductions to the datasets used in our experiments.

- **MathQA** (Amini et al., 2019): A dataset of math word problems requiring models to parse natural language into equations and perform multi-step numerical reasoning.
- **OpenBookQA** (Mihaylov et al., 2018): A multiple-choice QA benchmark where solving elementary science questions requires combining provided core facts with additional commonsense knowledge.
- **Winogrande** (Sakaguchi et al., 2020): A large-scale commonsense reasoning dataset for pronoun resolution, carefully constructed with adversarial filtering to reduce annotation artifacts.
- **BoolQ** (Clark et al., 2019): A yes/no question answering dataset with naturally occurring queries paired with passages, testing reading comprehension and judgmental reasoning.

Table 4 shows statics of datasets.

Table 4: Summary of Question Answer Dataset.

| Dataset | #Train | #Test |
|---|---|---|
| Mathqa | 30k | 3.0k |
| Boolq | 9.2k | 3.2k |
| Openbookqa | 5.0k | 0.5k |
| Winogrande | 9.2k | 1.3k |

## D EXPERIMENT DETAILS

### D.1 DETAILS OF GLUE

We add a classification layer on top of DeBERTaV3-base, with the output dimension for each task consistent with Table 3. The specific training hyperparameters of our method and baselines for each task are in Table 5. {} indicates that a grid search was performed while keeping the other hyperparameters fixed. Moreover, we ensure that all combinations across all methods can be trained properly, with the loss decreasing and the accuracy improving as expected. And we We report the average results over all combinations for each method.

Table 5: Training Details for Glue.

| Dataset | Epoch | Batchsize | Learning Rate | Warm Up | interval |
|---|---|---|---|---|---|
| CoLA | 10 | 128 | $5 \times 10^{-4}$ | $\{100, 200, 300, 400\}$ | $\{10, 30\}$ |
| SST-2 | 1 | 128 | $5 \times 10^{-4}$ | $\{100, 200, 300, 400\}$ | $\{10, 30\}$ |
| MNLI | 1 | 128 | $5 \times 10^{-4}$ | $\{400, 1000, 2000, 4000\}$ | $\{200, 400\}$ |
| QQP | 1 | 128 | $5 \times 10^{-4}$ | $\{500, 1000, 1500\}$ | $\{100, 200\}$ |
| QNLI | 1 | 128 | $5 \times 10^{-4}$ | $\{400, 700, 1000, 1300\}$ | $\{100, 200\}$ |

### D.2 DETAILS OF LLM TUNING AND EVALUATION

We conduct fine-tuning on three models across four datasets, with the training details summarized in Table 6.

Table 6: Training Details for Question Answer.

| Dataset | Epoch | Batchsize | Learning Rate | Warm Up | interval |
|---|---|---|---|---|---|
| Mathqa | 2 | 128 | $5 \times 10^{-5}$ | $\{100, 200, 300, 400\}$ | $\{10, 30\}$ |
| Boolq | 10 | 128 | $5 \times 10^{-5}$ | $\{100, 200, 300, 400\}$ | $\{10, 30\}$ |
| Openbookqa | 10 | 128 | $5 \times 10^{-5}$ | $\{100, 200, 300, 400\}$ | $\{10, 30\}$ |
| Winogrande | 10 | 128 | $5 \times 10^{-5}$ | $\{100, 200, 300, 400\}$ | $\{10, 30\}$ |

We fine-tune the LLM with standard instruction tuning, and during evaluation we compute the perplexity (PPL) of each candidate option and select the one with the lowest perplexity (i.e., highest probability) as the prediction.

## E THE USE OF LARGE LANGUAGE MODELS

We used a large language model (LLM) solely as a writing assistant to improve the clarity and readability of the manuscript (e.g., polishing grammar and phrasing). The LLM was not involved in research ideation, experimental design, implementation, or analysis. All scientific contributions and results are entirely the work of the authors.

