# OpenReview forum: "Dynamic Rank Reallocation for Module-Level LoRA"
_ICLR.cc/2026/Conference — ICLR 2026 Conference Withdrawn Submission_

### Official Review · Reviewer_tm1S · 2025-10-23

**Soundness:** 2
**Presentation:** 2
**Contribution:** 1
**Rating:** 2
**Confidence:** 4

**Summary:**

This paper proposes DaRA (Dynamic rank Re-Allocation), a module-level dynamic rank reallocation scheme for LoRA. The method warms up with a uniform rank, then decreases ranks for less important modules and increases them for critical ones under a fixed parameter budget. Technically, the authors rewrite LoRA as ΔW=BΛA with orthogonality regularization onA,B to decouple directions; the learnable diagonal Λ provides per-direction scaling. They introduce the PQ Index (PQI) to quantify the “density/sparsity” of Λ and to decide “how much to prune and where to re-allocate,” together with a theoretical lower bound and a pseudocode procedure. Experiments on GLUE (DeBERTaV3-base) and QA/reasoning tasks with OPT-125M, Qwen-1.8B, and LLaMA3.1-8B show modest but consistent gains over LoRA, AdaLoRA, and IncreLoRA; layer-module rank distributions and convergence curves suggest allocating more ranks to projection/FNN layers and to later-layer QKV.

**Strengths:**

- **Unified view of rank increase/decrease.** The paper argues that optimal ranks vary across modules and datasets; hence a fixed global rank is suboptimal. DaRA makes post-warm-up reallocation the core, unifying prior one-sided strategies (only decrease or only increase).
- **Simple and lightweight.** The BΛA parameterization adds only r scalars per module and uses orthogonality constraints; PQI is computed directly from Λ without extra gradients.
- **Reasonable empirical coverage.** Across GLUE and several QA/reasoning benchmarks, DaRA achieves parity or small gains under **comparable rank budgets,** with supportive analyses on rank distribution and convergence.

**Weaknesses:**

- **Limited novelty boundary.** Reallocation extends and combines prior dynamic-rank ideas, rank pruning first then rank increasing; the main innovations—BΛA and PQI-based selection—are natural extensions of factorization and sparsity measures. Please clarify why the specific combination of Λ+orthogonality+PQI is crucial beyond simpler importance heuristics.
- **Theory–practice gap.** The interference theory analogy is motivating but not analytically tied to the proposed method. “The authors provide a detailed schematic of the theory in Figure 3; however, it has not been effectively integrated with the proposed method. At present, the relationship between the two appears to be merely a simple sequential connection.
- **Figures under-explain key phenomena.** Figure 3 comprises four plots intended to illustrate how the Contribution and Rank of different modules evolve during fine-tuning. Unfortunately, panels (a) and (c) fail to reveal the anticipated variations, instead exhibiting a high degree of consistency. Meanwhile, the Rank changes highlighted in (b) and (d) are rather subtle, and no concrete examples are supplied to clarify their significance, which weakens the overall persuasiveness of the demonstration.
- **Lack of recent baselines.** The experiments are currently limited to comparisons with standard LoRA and a few dynamic-rank variants; they do not include state-of-the-art LoRA derivatives such as LoRA-Pro and LoRA-GA. These recent methods also aim to achieve superior performance under the same trainable-parameter budget, making them highly relevant baselines. The authors are encouraged to incorporate comparisons with these advanced variants to fully demonstrate the advantages of the proposed approach.

**Questions:**

- **What are DaRA_raw and DaRA_task, respectively?** The paper introduces two variants, DaRA_raw and DaRA_task, but does not clearly define their differences or respective roles.
- **What does the heatmap in Figure 5 represent? What is the meaning of each value?** The heatmap in Figure 5 is presented without sufficient explanation. Please specify what each cell value represents.
- **PQI Configuration and Ablation Study.** How are the hyperparameters *p*, *q*, and *η* selected in the PQI (Parametric Quantization Index) formulation? What is their impact on the lower bound of rank reduction and the final model accuracy? Moreover, could the authors provide sensitivity curves across different tasks and recommend default values for these parameters?
- The font size in Figures 1, 2, 3 and 5 is too small to read comfortably; please enlarge the fonts so that they match the size used in the captions.
- The caption of Table 2 is misleading: it claims the table compares training time, whereas the content actually reports model performance. Please revise the caption to reflect that it is a performance comparison across different models.

---

### Official Review · Reviewer_aXq5 · 2025-10-30

**Soundness:** 2
**Presentation:** 2
**Contribution:** 2
**Rating:** 2
**Confidence:** 4

**Summary:**

The paper addresses a key limitation in standard LoRA: the static, uniform rank allocation across all model modules, which fails to capture the heterogeneous importance of different layers for a specific downstream task.

The authors propose the **Dynamic rank re-Allocation method (DaRA)**, a novel approach inspired by the **interference theory in human learning** (forgetting outdated knowledge to facilitate new acquisition). DaRA implements a two-stage dynamic process:

- **Warm-up:** Initial uniform rank allocation provides a coarse understanding of the task.
- **Reallocation:** Ranks are dynamically adjusted by unifying both **rank decrease** (discarding less useful directions from unimportant modules) and **rank increase** (assigning new capacity to critical modules).

This dynamic mechanism is enabled by:

- A reformulated LoRA update $\Delta W=B\Lambda A$, where $\Lambda$ is a learnable diagonal scaling matrix.
- An orthogonality regularization on $A$ and $B$ to ensure each rank-1 component corresponds to a distinct direction in the parameter space.
- The **PQ Index (PQI)**, a lightweight metric, is applied to $\Lambda$ to efficiently assess module importance and guide the rank reallocation process.

**Strengths:**

The primary strength of DaRA lies in its **holistic and unified approach to dynamic rank management**.

- **Pioneering Unified Reallocation:** DaRA is one of the first methods to effectively unify both rank reduction (like AdaLoRA) and rank expansion (like IncreLoRA) into a single, seamless reallocation paradigm. This grants superior flexibility, ensuring that the model's fixed parameter budget is maximally utilized by dynamically transferring capacity from less-important to more-important modules during training.
- **Strong Empirical Performance:** The method demonstrates consistent performance improvements over existing state-of-the-art PEFT baselines (LoRA, AdaLoRA, IncreLoRA) across a diverse range of models  and tasks  under comparable parameter budgets.
- **Biologically Inspired Justification:** The explicit analogy to cognitive interference theory provides an intuitive and theoretically appealing framework for the forget-and-reinforce learning mechanism implemented by the dynamic rank adjustment.
- **Efficiency:** The reliance on the lightweight PQ Index metric for module importance allows for rank decisions to be made efficiently.

**Weaknesses:**

- Sensitivity to Scheduling Hyperparameters: The performance of any dynamic-schedule method is highly sensitive to when rank changes occur. The reliance on predefined **warm-up steps** and **regular intervals $\nu$** for rank expansion introduces critical hyperparameters that must be carefully tuned per-task. A suboptimal schedule can lead to premature pruning (catastrophic forgetting) or delayed capacity increase, hindering the fine-tuning process. This adds complexity compared to the fixed-rank LoRA baseline.
- DaRA’s core theory is based on rank reduction and expansion but ablation experiments are not provided. If merely allowing rank increases, without rank reduction, could achieve similar performance, then the theoretical and practical value of the overhead and complexity introduced by DaRA’s rank reduction and complex orthogonality regularization mechanisms would be questionable.
- **While the average performance of this method shows an improvement of approximately 0.5% on GLUE tasks and 0.1%–0.2% on QA tasks compared to AdaLoRA and IncreLoRA, the statistical significance of this improvement has not yet been validated.** Given the marginal performance differences, these gains may simply be noise resulting from variations in random seeds, hyperparameter search bias, or data batch ordering. **The paper must provide rigorous quantitative support for its superiority claims by conducting multiple randomized repeat experiments to compute standard deviations or confidence intervals, and by using statistical methods  to prove that the performance gains are statistically significant.** Otherwise, the claims of superiority lack rigorous quantitative support.
- The authors observe a trend: a propensity to reduce the rank of QKV modules while increasing the rank of projection layers and FFN modules. However, this final allocation trend may be highly dependent on the initial rank configuration ($\mathbf{R}_{initial}$) at the start of training. The paper needs to provide ablation studies with different initial rank configurations to prove that this inter-module rank transfer trend (QKV↓, FFN↑) is a general, task-driven phenomenon, rather than an artifact related to the initial setup.

**Questions:**

1. **Does setting different initial ranks yield approximately the same final rank?** The authors should provide ablation studies to demonstrate the **robustness and convergence** of the method. Specifically, under the same total rank budget, the paper should test at least two distinct $\mathbf{R}_{initial}$ configurations (e.g., one high-rank configuration and one low-rank configuration) and present a comparison of the final module rank allocation ($\mathbf{R}_{final}$) results.
    - **If the $\mathbf{R}_{final}$ is similar:** This would strongly suggest that DaRA can find a general, task-relevant rank distribution, thus confirming the effectiveness of the PQ Index.
    - **If the $\mathbf{R}_{final}$ differs significantly:** This would indicate that the method is highly dependent on the initial hyperparameter configuration, thereby weakening its value as a general dynamic allocation method.
2. Does the value of the PQ Index remain stable or fluctuate drastically during training? How does the value of the PQ Index change when the rank is reduced or increased? The paper needs to provide a curve plot illustrating the change of the PQ Index over training steps to demonstrate its reliability

---

### Official Review · Reviewer_umTN · 2025-10-31

**Soundness:** 2
**Presentation:** 3
**Contribution:** 2
**Rating:** 4
**Confidence:** 3

**Summary:**

This paper introduces DaRA (Dynamic rank re-Allocation), a novel method for improving LoRA. The core motivation is that existing LoRA-based methods typically assign a uniform, fixed rank to all modules, which is suboptimal. DaRA dynamically reallocates ranks during training, starting with a uniform rank allocation during a warm-up phase, then uses a lightweight metric, the PQ Index (PQI), to identify important and unimportant modules. The experiments demonstrate that DaRA consistently outperforms existing baselines like LoRA, AdaLoRA, and IncreLoRA under similar parameter budgets.

**Strengths:**

1.The paper convincingly argues that a dynamic, bidirectional rank adjustment is superior to static or unidirectional (increase-only or decrease-only) approaches. The proposed method, which unifies rank pruning and rank growth into a single "reallocation" framework, is a logical and powerful extension
2.The core modification to LoRA is elegant. It introduces only one learnable scalar per rank-1 component, and the use of the PQI metric is claimed to be a lightweight way to guide the reallocation process. This makes the method easy to integrate into existing LoRA implementations.
3.The empirical evaluation is thorough, covering both encoder and decoder architectures of varying scales on a diverse set of NLU and QA tasks. The paper also provides valuable analyses, such as the rank distribution visualization and convergence speed comparison.

**Weaknesses:**

1.Insufficient Ablation Studies: This is the most significant weakness. The paper introduces several new components (the learnable \lambda scalars, the orthogonal regularization term, and the PQI metric), but their individual contributions are not isolated.
2.A comparison of the extra training time, FLOPs, or memory overhead against baselines is missed.
3.Minor Presentation Issues: While the paper is generally well-written, there are several typos and minor clarity issues that should be addressed in the final version:
a)The model name seems to be incorrect in lines 406-407.
b)Table 1 header: "Parmas" should be "Params".
c)Algorithm 1: "initalization" should be "initialization".

**Questions:**

1.The definition of the PQ Index (Eq. 10) depends on hyperparameters p and q. What values were used for p and q in the experiments? How sensitive is the model's performance to the choice of these hyperparameters?
2.To support the claim of efficiency, could you provide a quantitative comparison of the computational cost (e.g., wall-clock time per training step) of calculating the PQI scores versus the importance scores used in AdaLoRA and IncreLoRA under a controlled experimental setup?

---

### Official Review · Reviewer_BGHC · 2025-11-01

**Soundness:** 3
**Presentation:** 3
**Contribution:** 2
**Rating:** 4
**Confidence:** 4

**Summary:**

The authors propose DaRA (Dynamic rank re-Allocation), a new PEFT method addressing the fixed rank allocation in LoRA. While existing methods like AdaLoRA only prune ranks and IncreLoRA only adds them, DaRA effectively does both. The approach uses a two-stage process: a uniform-rank "warm-up" phase, followed by dynamic reallocation guided by a "PQ Index" (PQI). This metric, from pruning literature, helps determine which ranks to prune and which to reinforce, a process likened to human learning. Experiments on GLUE, MathQA, and other benchmarks show DaRA outperforms baselines.

**Strengths:**

1.  **Clear Motivation:** The paper provides a strong motivation for the work. The analysis in Section 3 (Figs. 1 & 2) gives a clear, data-driven reason why a uniform rank allocation is suboptimal, as the authors effectively demonstrate the heterogeneous rank requirements of different modules.
2.  **Sound Method:** The core methodology of unifying rank pruning and growing is sound. Using the PQI metric as a lightweight guide is an effective choice, as it avoids the heavy computational costs of other importance-scoring methods.
3.  **Valuable Analysis:** The additional analysis in Section 6 is valuable. The plots showing the PQI correlation with performance (Fig. 4) and the convergence speed comparison (Fig. 6) offer insights into the method's behavior.

**Weaknesses:**

1.  **Insufficient Justification for PQI:** The justification for using PQI is less comprehensive for *increasing* rank. The paper uses Theorem 1 to support *pruning*, but the logic for using high PQI to *add* rank appears to be based more on intuition ("potential benefit") and empirical correlation rather than strong theoretical backing.
2.  **Lack of Key Ablation Studies:** The method introduces two main changes: the $B \Lambda A$ parameterization and the PQI-guided dynamic allocation. The experiments do not decouple these, making it difficult to determine the individual performance contribution from the new structure versus the dynamic reallocation strategy.
3.  **Unclear Tuning Costs:** DaRA introduces new hyperparameters, including warm-up steps $W$ and update interval $\mu$. While a grid search is mentioned in the appendix, no sensitivity analysis is presented. This makes it difficult to assess the method's robustness and practical tuning cost.
4.  **Unsupported 'Lightweight' Claim:** The paper repeatedly claims the method is "lightweight" and computationally cheap, but this is not substantiated with quantitative data. There are no comparisons of wall-clock training time or VRAM usage against the baselines. (Table 2's title also incorrectly says "Time" when it's showing "Acc"). This makes an assessment of its practical utility difficult.

**Questions:**

See weaknesses.

---

### Note · Authors · 2025-11-19

I have read and agree with the venue's withdrawal policy on behalf of myself and my co-authors.